# Method for Manufacturing Corn Straw Cement-Based Composite and Its Physical Properties

**DOI:** 10.3390/ma15093199

**Published:** 2022-04-28

**Authors:** Boyu Niu, Byeong Hwa Kim

**Affiliations:** 1Department of Civil Engineering, Beihua University, No. 3999, Binjiang East Road, Jilin 132013, China; niuboyu@163.com; 2Department of Civil Engineering, Kyungnam University, Changwon 51767, Korea

**Keywords:** corn straw, rural housing, building energy efficiency, ordinary Portland cement

## Abstract

This paper introduces an innovative method for making cement-based composites from corn straw plants, and investigates the strength, thermal conductivity, and hydration characteristics of the composites. Corn straw is a natural, renewable, and breathable thermal insulation composite that contains cellular sealed pores. Corn straw contains a large amount of soluble cellulosic sugar, which hinders the hydration reaction of Portland cement and affects the use of corn straw as a building material. In this study, a 3 wt.% siliceous solution was used for surface treatment of corn straw particles to prevent cellulosic sugar from affecting the hydration performance of Portland cement. The composition of added cement-based composite materials with treated corn straw at the dosage of 11–20 wt.% was investigated. The test results showed that the corn straw cement-based composite (CSCC) had an optimal thermal conductivity of 0.102–0.112 (W/(m·K)) and a minimum compressive strength of above 1 MPa. The hydration performance of four typical CSCCs was examined using XRD, SEM, and EDS. The experimental results of this study may help to increase the comprehensive utilization of corn straw. The manufacturing method of the composite materials is simple, effective, and convenient for popularization and application, and it provides a new important technical measure to solve the problem of high energy consumption in rural houses.

## 1. Introduction

While the rapid economic development of human society has continuously improved people’s living standards, a number of energy and environmental problems have also emerged. Seen from the structure of energy consumption in China, building energy consumption accounts for 46.7% of total energy consumption, of which 61.4% takes place in rural areas [1,2]. Many scholars in the world agree that external wall insulation is the most efficient technique for saving energy in a building. Characterized by simple operation, sound effect, low investment, and high feasibility, external wall insulation has become a major building energy conservation technique [3,4,5,6].

China is rich in straw resources. The yield of agricultural residue corn straw is huge in China. Approximately 32.8% of corn straw is yielded in the cold areas of northern China [7]. Affected by geography, climate, economy, science, and technology, most corn straw is burned on site, which seriously pollutes the atmosphere, harms human health, and poses a challenge to production and life [8].

Traditional cement-based composite materials blocks have a thermal conductivity of 0.92~1.32 (W/m·K), which means a high building cost, considerable energy consumption, and heavy dependence on non-renewable resources. This makes them unsuitable for long-term human development [9]. Agricultural and forestry waste, as a type of building material, has been in use for thousands of years [10]. In modern developed countries, approximately 40% of agricultural and forestry resources are consumed by the building industry. The role of agricultural and forestry waste as a supplementary material in the building industry is increasingly recognized [11,12]. According to statistics from the past decade, straw can reduce at least 40% of CO_2_ emissions compared to traditional building materials [13]. Studies [14,15,16,17,18] have shown that agricultural and forestry waste, as building wall material, has the following advantages: (1) Agricultural and forestry wastes is generally characterized by low density, low compressive strength, small elastic modulus, and high tensile strength; (2) Most agricultural and forestry waste contains numerous sealed pores, which show excellent thermal insulation and acoustic insulation performance; (3) Agricultural and forestry waste is a cheap renewable resource that can be easily obtained. However, there is a wide variety of agricultural and forestry waste, and the above studies rarely pay attention to corn straw as a building wall material.

Corn straw contains large amounts of cellulose, hemicellulose, and lignin. Among them, cellulose and hemicellulose hydrolyze in water and release monosaccharides, such as glucose, xylose, arabinose, and galactose, which are collectively called cellulosic sugar [19,20]. Xie [21] discovered that untreated agricultural and forestry waste contains a large amount of cellulosic sugar, which affects the hydration reaction of Portland cement and the use of plant fibers in cement-based composite. Wang [22] found that 46.88 wt.% cellulosic sugar can be extracted from 15 g corn straw by hydrolysis. Literature [23] confirmed that cellulosic sugar prolongs the setting and hardening time of cement. Moreover, cellulosic polysaccharides of corn straw are sensitive to alkaline corrosion and can be easily dissolved [24]. When the dosage of cellulosic sugar exceeds 0.1 wt.%, the compressive strength of cement-based composite decreases [25]. Excess cellulosic sugar also prolongs the hydration time of cement and hinders the molding of C3S crystal [26]. The literature has revealed that NaOH solution offers the most effective way to remove cellulosic sugar [27]. However, treated straw needs to be cleared several times with clean water to prevent NaOH solution from causing alkaline corrosion of cement-based composites, which causes high costs [9,21,28]. At the same time, Nassar [29] found that adding hydroxypropyl methylcellulose (HPMC) into cement slurry can obtain a uniform plant fiber cement-based mixture. The Kyong [30] team found that different doses of natural plant fiber have a certain impact on the compressive strength of plant fiber cement materials.

In this study, corn straw was treated with siliceous solution, which weakens the water absorbency of corn straw, avoids hydrolysis of cellulosic sugar, prevents corn straw from affecting the hydration performance of Portland cement, and circumvents alkaline corrosion caused by the hydration reaction between Portland cement and water. Effects of various doses of treated corn straw on the compressive strength, thermal conductivity, and hydration heat of cement-based materials were analyzed. This study addresses two issues. One is an innovative recipe first introduced to make corn straw cement-based composites, which are easy to popularize and produce. The other is that it is the first experimental study of the various physical properties of corn straw cement-based composites. This approach can increase the overall utilization of corn straw at low cost without causing secondary pollution, reduce the dependence on natural resources for building materials, improve thermal insulation and energy saving of buildings in rural areas of northern China, and solve the problem of excessive energy consumption.

## 2. Materials and Methods

### 2.1. Materials and Mixes

Dinglu 32.5 ordinary Portland cement (OPC) produced by Yatai Group (Jilin Province) has a 28 days compressive strength of 39.82 MPa, a flexural strength of 6.89 MPa, and an active slag dosage of 7.56 wt.%. Other mineral compositions of the product are given in Table 1.

The corn straw was collected in Jilin Province, China. After air drying, it was ground with an electric hay cutter, as illustrated in Figure 1. The size of corn straw was controlled at 1.5–2 cm and dried in the oven at 85 °C for 24 h to remove excess moisture. The main dissolved components of corn straw were tested (Table 2).

Siliceous solution is an alkaline aqueous solution obtained by the reaction of potassium hydroxide and methylsiliconic acid. It can react with CO_2_ or other acidic substances in the air to create poly-methylsiliconic acid, producing a waterproof performance. The main components of the siliceous solution are shown in Table 3.

Hydroxypropyl methylcellulose (HPMC) is a cement paste thickener that can prevent porous materials from absorbing moisture in Portland cement paste and protect Portland cement paste from segregation and bleeding [31]. The HPMC used in this experiment was white, with a viscosity of 3000 mPa.s and a pH of 7.0–9.0. Figure 2 illustrates the molecular structure of HPMC. Relying on supercritical binding between Ca^2+^ and the ether group (R-O-R), Portland cement particles can be adsorbed to HPMC molecules, leading to hydrogen-bonded molecular cross-linking. This improves the cohesion and water retention of Portland cement paste and modifies its fluidity [32]. HPMC also increases the homogeneity of corn straw as a cement-based composites aggregate and ensures a consistent thermal insulation of corn straw and uniform quality of target products.

Other materials such as superplasticizer, glass fiber, and other raw materials are not investigated here.

### 2.2. CSCC Production Method

In this experiment, the mix ratio of 1 m^3^ corn straw cement-based composite material is used as the basis of the mixing ratio design. Dosages of different ingredients are shown in Table 4.

In this experiment, siliceous solving liquid was used for surface treatment of corn straw. Siliceous solution and water were mixed in the ratios of 1 wt.%, 2 wt.%, 3 wt.%, 4 wt.%, and 5 wt.%. The mass ratio between mixing water and corn was 120 wt.%. Corn straw and siliceous solution were mixed with a mixer. Relying on the water absorbency of corn straw, siliceous solution was naturally absorbed into corn straw to avoid secondary pollution. The treated corn straw was dried at 80 °C for 24 h. The dried corn straw is mixed with cement to make a dry corn straw cement mixture. The mixing time is 1 min. After the water reducer and glass fiber are mixed with water, siliceous solution and HMPC are added and mixed to make mixing water. The mixing water is evenly mixed with the corn straw mixture to produce CSCC slurry. For curing, the slurry is placed into the mold (100 × 100 × 100 mm^3^). The removal time from the abrasive tool depends on the hydration and formation state of the CSCC. The specific steps are shown in Figure 3.

### 2.3. Experimental Procedure

The experimental procedure is shown in Figure 4.

#### 2.3.1. Physics Experiments

Compressive strength is tested in cube shape. Samples are taken on days 3, 7, 14, 28, and 56. Cube size is 100 × 100 × 100 (mm^3^), and the dimensional error is less than 2 mm. Compressive strength ‘R_p_’ (MPa) is calculated as R_p_ = F/A. Here, ‘F’ and ‘A’ represent the maximum failure load (N) and the area of the pressure surface (mm^2^), respectively.

Compressive strength is tested in cube shape. Samples are taken on days 3, 7, 14, 28, and 56. Cube size is 100 × 100 × 100 (mm^3^), and the dimensional error is less than 2 mm. Compressive strength ‘R_p_’ (MPa) is calculated as R_p_ = F/A. Here, ‘F’ and ‘A’ represent the maximum failure load (N) and the area of the pressure surface (mm^2^), respectively.

The thermal conductivity test method adopts GB/T--32981. The cold box temperature is set to −15 °C, and the hot box temperature is set to 25 °C. Sample size is 300 × 300 × 30 (mm^3^). All specimens were tested after 56 days of natural curing. The calculation is given in Equation (1).
(1)$$\lambda_{\varepsilon }=\frac{Q\times d}{S\times \left(T_{2}-T_{1}\right)}\times k$$
where
λ_ε_ = thermal conductivity of sample (W/m·K).Q = sum sample heat flow rate (W).T_1_ = average temperature of cold box (°C).T_2_ = average temperature of metering box (°C).S = sample area (m^2^).d = sample thickness (m).k = correction factor is 1.03.


Instrument model ITC-200 of Isothermal Titration Calorimetry is used for this test. A sample of 10 g of CSCC was taken on site and placed in the instrument within 10 min. The computer records the heat change over 160 h and automatically collects the calculation results.

#### 2.3.2. Analytical Method

For the X-ray powder crystal diffraction experiment, D8-Focus manufactured by AXS company in Germany was used. A sample of 50 g was taken and ground with a ceramic ball mill. The scanning speed was set to 4°, 2θ/min. The test results were analyzed with MDI jade 6.

A model Mira 3 equipped with energy X-ray spectrum detector (EDS) of a scanning electron microscope manufactured by TESAN was used. The vacuum negative pressure value of the test sample was set to 50 Pa at voltage 15 kV and a current of 8 to 10 mA. A relatively flat plane was adopted as the scanning plane. For the accuracy of the test data, the cement blocks were immersed in anhydrous high-purity ethanol and sealed to terminate the hydration of the cement. Before testing, the specimens were dried in an oven at 70 °C for 2 h and sprayed with metal.

## 3. Results and Discussion

### 3.1. Performance Analysis of Corn Straw

Corn straw contains numerous cellular sealed pores and is a natural, breathable thermal insulation material. The structural layer of corn straw is composed of cellulose, hemicellulose, and lignin. Corn straw has an extremely high waters absorbency, which can reach 140 wt.% within 24 h (Table 5). Due to the poor stability of substances such as cellulosic sugar and their sensitivity to alkaline substances, they are freely soluble in alkaline liquids and have a delayed coagulation effect on Portland cement [34]. The team of Lei [25] found that when the content of plant cellulosic polysaccharides is larger than 0.1 wt.%, C3S and C3A in the hydration reaction of Portland cement are inhibited 16 h before hardening. As the dosage of cellulosic sugar increases, the initial setting time and final setting time of Portland cement are prolonged, followed by a decrease in hydration heat and compressive strength, and even termination of hydration in Portland cement [35]. The team of Savastano [36] discovered that pretreated straw can improve the adhesion between plant fibers and Portland cement and increase the resistance of plant fibers to corrosion by alkaline solution. The team of Li [37] performed surface treatment of timber using SiO_2_ particles and discovered that the treated timber is characterized by high hydrophobicity and obviously improved resistance to acids and alkalis.

In this experiment, siliceous solving liquid was used for surface treatment of corn straw. The siliceous solution would react with CO_2_ in the air, creating a breathable meshed waterproof membrane on the surface of the treated corn straw. In that case, the treated corn straw would have excellent hydrophobicity without changing its original natural structure or thermal insulation. Test and comparison results indicated that the best effect was achieved when the concentration of siliceous solution was 3 wt.%, in which case the angle between the water drops and corn straw was less than 90° (Figure 5). We know that the main chemical components of corn straw consist of lignin, hemicellulose, and cellulose, the last of which is the primary structural component of the cytoderm [38]. In this experiment, untreated corn straw (S1) and treated corn straw (S2) were analyzed by XRD (scanning range: 10° ≤ 2θ ≤ 30°), as shown in Figure 6. It is clear that the diffraction peak of untreated corn straw appeared near 16.3°, whereas the peak of treated corn straw appeared near 16.9°. The diffraction peak of the cellulose appeared near 22.7° in both cases. A trough near 18.9°, i.e., amorphous area diffraction, was observed. The treated corn straw had a higher diffraction peak, which fully reflected the change in the composition of cellulose crystals after the treatment of the corn straw. This proved that siliceous solution to some extent changed the crystallinity of cellulose in the corn straw, reduced amorphous components in the fibers and partially removed cellulosic sugar [38]. Bilba [39] found that Si^+^ deposited on the cytoderm of the plant fiber gave it a better interface profile under SEM, as shown in Figure 7. The treated corn straw showed ultrastructure of plant fibers. The cytoderm was complete with a clear profile and an intact cavity, and the waterproof structure took shape. Free water could not penetrate the corn straw, and cellulosic sugar could not hydrolyze, so cellulosic sugar could not hinder the hydration reaction of Portland cement. The basic properties of the treated corn straw are shown in Table 5.

### 3.2. Performance Analysis of CSCC

The results of compressive strength and thermal conductivity of CSCC are shown in Table 6 and Figure 8, respectively. As a result of the tests, the treated corn straw served as an aggregate in cement-based composites. At low dosages, the strength growth of CSCC was similar to that of ordinary cement-based composites. However, as the mass of treated corn straw increased, the compressive strength of the cement-based composites decreased uniformly with age.

According to the literature [40], plant straw itself has low compressive strength and elastic modulus. When used as coarse aggregate in a cement-based composite, it has no skeleton effect and greatly reduces the density and compressive strength of the cement-based composite. As shown in Figure 8b, when the corn straw parameters are 11 wt.%~15 wt.%, the compressive strength of CSCC is 0.9 MPa and 0.41 MPa in 3 days and 6.72 MPa and 4.9 MPa in 56 days. Existing literature has shown that plant straw particle size, water content, and aggregate gradation affect the thermal conductivity and compressive strength of corn straw [41]. As shown in Figure 9, when the amount of corn straw is 46.2 kg (CSCC-11 wt.%), the thermal conductivity of CSCC is 0.201 (W/(m·k)), and the compressive strength is 6.72 MPa. However, With the increase in the amount of corn straw, the thermal conductivity and compressive strength of CSCC began to decline. When the quantity increases to 79.8 kg (CSCC-19 wt.%) and 84 kg (CSCC-20 wt.%), the compressive strength of the material continues to decline (compressive strength is 0.56 MPa and 0.44 MPa), whereas the thermal conductivity does not further decrease and is basically stable at about 0.099 (W/(m·k)). The thermal conductivity of ordinary cement-based composite wall material is 0.92~1.32 (W/(m·k)) [9], indicating that CSCC is a very excellent building energy-saving wall material.

### 3.3. HPMC Mission in CSCC

HPMC is a cement paste thickener. Relying on supercritical binding between Ca^2+^ and the ether group (R-O-R), Portland cement particles can be adsorbed to HPMC molecules, leading to hydrogen-bonded molecular cross-linking. This improves the cohesion and water retention of Portland cement paste and modifies its fluidity [31]. HPMC also increases the homogeneity of corn straw as a cement-based composite aggregate and ensures a consistent thermal insulation of corn straw and uniform quality of target products. As can be seen from the CSCC slice in Figure 10, the apparent density of corn straw is lower than that of cement paste, so when mixed together, the corn straw is subjected to vibration and floats on the surface of the cement-based composite, resulting in non-uniform mass distribution in CSCC. Therefore, adding HPMC to the CSCC creates air bubbles within the CSCC but produces a thickening effect. However, HPMC prevented upper floatation and bubbling of corn straw and prevented delamination and segregation in CSCC. The air bubbles had some effect on the compressive strength of the CSCC, but the sealed bubbles reduced the thermal conductivity of the material.

### 3.4. Effect of Corn Straw Dose on Heat of Hydration

When the amount of corn straw increased to 19 wt.% and 20 wt.%, the compressive strength of the cement-based composite decreased significantly, and the compressive strength was 0 MPa in 3 days, even if the compressive strength was only 0.56 MPa and 0.44 MPa after 56 days. This shows that the hardening of cement also depends on the concentration of cellulose sugar extracted from plant fibers. If the concentration of cellulosic sugar was ≥0.1 wt.%, the setting time of the cement would be extended [9]. In general, the addition of 10 wt.% straw in the cement-based composite causes the hydration heat rate of cement to approach zero in 7 days [21]. The effect of hydration heat of treated corn straw in cement-based composite was tested by isothermal titration calorimetry. Figure 11 and Table 7 show the changes of heat flow rate (mW/g) and heat of hydration (J/g) of CSCC within 168 h under constant cement mass when the dosage of treated corn straw fell in the range of 15 wt.%~20 wt.%. The peak heat flow rate of pure ordinary Portland cement appears at 22.17 h, the peak heat flow rate is 2.20 (mW/g), and the heat of hydration is 375.6 (J/g). When the amount of treated corn straw was 15 wt.%~20 wt.%, the peak heat flow rate appeared between 37.96 h and 84 h, the peak heat flow rate decreased from 1.91 (mW/g) to 1.30 (mW/g), and the accumulated heat of hydration decreased from 316.4 (J/g) to 116.2 (J/g) linearly. This shows that the heat flow rate and heat of hydration of cement decrease with the increase in corn straw. As shown in Figure 11b, when the dosage of corn straw is 19%, the hydration heat of CSCC decreases to 1/2 of that of pure ordinary Portland cement, and the peak heat flow is delayed to 78.83 (hours), but the hydration reaction of cement did not stop. According to the literature [25,26], this is mainly because, after hydrolysis and migration of cellulosic sugar, it was adsorbed on Ca^2+^ products, which prevented the C3S hydrate from turning into C-S-H. As a result, it affected the cement hydration process, exerted a delayed coagulation effect, and postponed the occurrence of the maximum hydration heat. According to the test results, increasing the dosage of corn straw reduces the hydration capacity of Portland cement and the strength of set cement, which testifies to the limited hindering effect of siliceous solution.

### 3.5. Mechanism Analysis of CSCC

As can be seen from the previous experiments, the dosage of added treated corn straw affected the hydration of the cement. A lower dosage of treated corn straw in CSCC would lead to higher thermal conductivity and density, in which case the hydration reaction of CSCC would be similar to that of ordinary cement. To better understand the effect of treated corn straw on the hydration of Portland cement, XRD analysis was performed on the hydration products of CSCC-5, CSCC-6, CSCC-7, and CSCC-8 on the 56 days (scanning range: 20° ≤ 2θ ≤ 50°) (Figure 12). Seen from XRD patterns, CSCC-7 and CSCC-8 showed an obvious gypsum crystal diffraction peak at 2θ = 20.7°. Compared to the other two materials, CSCC-7 and CSCC-8 Portland stones showed an obviously weakened intensities of diffraction peaks. This indicated that adding the treated corn straw in an excessively high dosage would lead to an incomplete reaction of tricalcium aluminate in gypsum and cement clinker. The diffraction peak of quartz crystal appeared at 2θ = 26.64°. The primary component of quartz crystal is SiO_2_. The diffraction peak of Portlandite crystals appeared at 2θ = 24.1°, 29.4°, 36°, 39.42°, and 43.18°, and the peak value decreased with increasing dosage of corn straw. This meant that the increased cumulative content of cellulosic sugar in corn straw began to hinder the hydration reaction of Portland cement, reduce the output of Ca(OH_2_), and weaken the hydration reaction of cement.

Fifty-six day crushed cement blocks were selected. The micromorphology of CSCC-5, CSCC-6, CSCC-7, and CSCC-8 was observed with a scanning electron microscope. Extremely high alkalinity in the hydration process of cement paste destroyed the stability of plant fibers, resulting in loss of adhesion between plant fibers and the cement paste [42]. As shown in Figure 13, the treated corn straw is uniformly wrapped by the cement hydration product, with a uniform hydration structure and clearly visible hydration interface boundaries. There was little difference between the hydration products of cement and those of ordinary cement-based composite. The treated corn straw was smooth and dense on the surface, without traces of acidic or alkaline corrosion. The cellular straw cytoderm inside corn straw did not experience compressive deformation. This suggests that the siliceous solution and CO_2_ can react with each other to form dense siliceous oxides, preventing corn straw cellulose from releasing hydrolyzed cellulosic sugar and other substances in alkaline environments. It also confirms the conclusions of Savastano [36] that waterproofing agents improve the adhesion between natural plant fibers and cement and enhance their thermal stability.

CSCC-5, CSCC-6, CSCC-7, and CSCC-8 were observed by TESCAN MIRA 3 EDS scanning electron microscope. Seen from the SEM images presented in Figure 14a, Figure 15a, Figure 16a and Figure 17a, EDS analysis was performed on all specimens using the interface between treated corn straw and Portland cement, estimating thus the distribution of the main elements. As can be seen from the full EDS spectrum analysis presented in Figure 14, Figure 15, Figure 16 and Figure 17, crystal particles containing several elements formed after CSCC hydration and distribution of Portland cement hydration products were identified. The main elements of the CSCC surface are O Kα1, C Kα1, Ca Kα1, Si Kα1, Al Kα1, and a small amount of Fe Kα1. Fe Kα1 is not found in EDS plane scanning, but a small amount can be found in the X-ray energy spectrum. This is because Fe Kα1 is wrapped by hydration products (C-S-H) and sinks to bottom of the surface.

As can be seen from Figure 14d, Figure 15d, Figure 16d and Figure 17d, only small amounts of Ca Kα1 and Si Kα1 were observed on the surface of treated corn straw. This proves that siliceous solution can effectively prevent the alkali corrosion of Portland cement to corn straw and prevent calcium hydroxide (Ca (OH_2_)) from weakening the structure of corn straw. The hydration products of Portland cement prevent calcification of plant fibers of corn straw so that they do not deposit on the dimensional surface of corn straw. The durability of corn straw plant fibers in cement stone is improved. At the same time, the cellulose sugar in corn straw does not affect the hydration process of cement.

## 4. Conclusions

In this study, ten mixing ratios were designed. The effects of various doses of treated corn straw on the compressive strength, thermal conductivity, and hydration heat of cement-based materials were analyzed. Four representative samples were determined and the microstructure was analyzed to observe the interfacial structure of corn straw and cement. At the same time, the thermal conductivity of CSCC is low due to a large number of closed holes and low bulk density in corn straw. In general, materials with a thermal conductivity of 0.25 (W/(m·K)) or less become insulating materials [43,44]. As shown in Table 6, the thermal conductivity of all CSCCs is 0.25 (W/(m·K)) or less, so insulation performance is good. Therefore, corn straw can be widely used in the field of insulation building materials and plays an important energy saving role in the field of new building wall materials in rural areas.

(1)In this study, a 3 wt.% siliceous solution was used for surface treatment of corn straw. Treated corn straw is converted from a hydrophilic material to a hydrophobic material through a change in its water absorbency. This approach has effectively solved the problem of difficulty or termination of hydration of Portland cement due to easy hydrolysis of polysaccharide substances in alkaline solution.(2)Relying on the binding between Ca^2+^ and the ether group (R-O-R), Portland cement particles can be adsorbed to HPMC molecules, leading to hydrogen-bonded molecular cross-linking. This improves the cohesion and water retention of Portland cement paste. HPMC also avoids delamination between corn straw and cement paste due to vibration, guarantees uniform distribution of corn straw in cement-based composite, and guarantees stable thermal conductivity of materials and the uniform quality of products.(3)Seen from the results of EDS analysis, the hydration interface between corn straw and Portland cement was obvious after treatment. The Portland cement hydration crystals closely intersected, and the Portland cement product C-S-H crystal was similar to ordinary cement-based composite. However, when the dosage of corn straw exceeded 19 wt.% of cement mass, Portland cement showed an obvious decrease in hydration strength, which suggests that the siliceous solution has limitations when it comes to the treatment of corn straw.(4)When corn straw was added to the cement-based composite, the bulk corn straw replaced a considerable volume of cement-based composite, so the effect of weight reduction was obvious. However, corn straw did not play a skeleton role in the cement-based composite, so the compressive strength of cement-based composite decreased with increasing dosage of corn straw. When the dosage of corn straw was 18 wt.%, the thermal conductivity of CSCC was 0.102 (W/(m·K)), and its compressive strength exceeded 1 MPa, which met the application requirements of light thermal insulation materials for frame structures.

One challenge encountered throughout this experimental study is the size distribution of the corn straw. At present, we can only control the size of the hay cutter 1.5–2 cm. The corn straw pulp, corn straw skin, and corn straw leaves differ in density and strength. Corn straw is very difficult to sort. Due to the large number of samples required for the experiment, an effective separation method has been yet found. This issue will be resolved in the near future.

## 5. Patents

Method for making corn straw plant fiber insulating masonry material; China. CN 201610546943.3. 11 September 2018.

## Figures and Tables

**Figure 1 materials-15-03199-f001:**
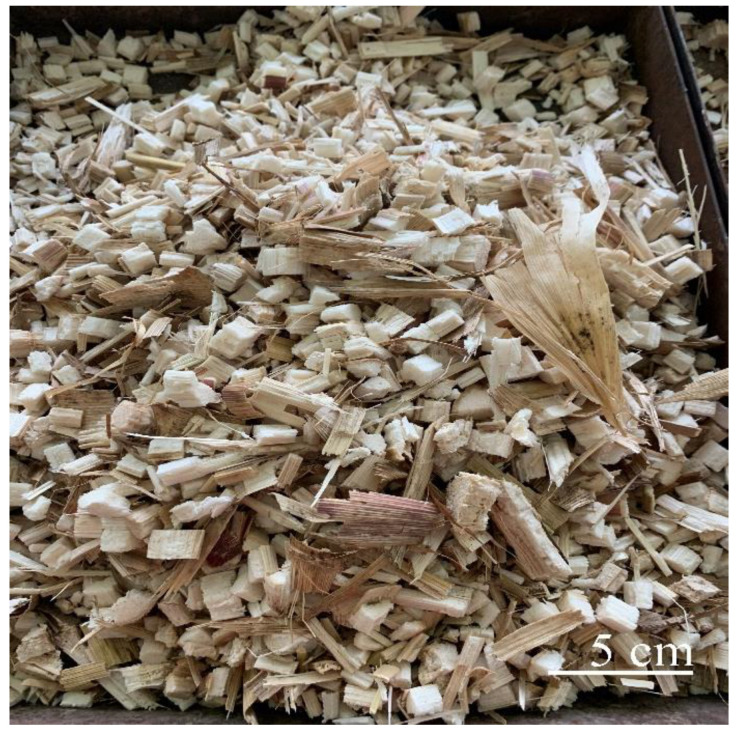
Crushed corn stover pellets.

**Figure 2 materials-15-03199-f002:**
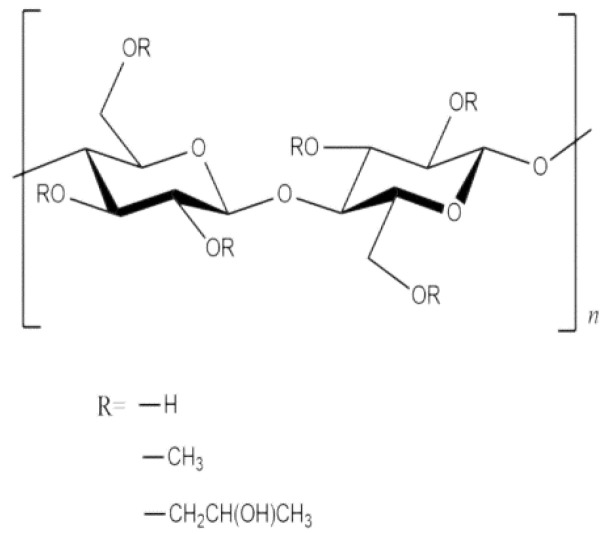
HPMC molecular structure diagram [33].

**Figure 3 materials-15-03199-f003:**
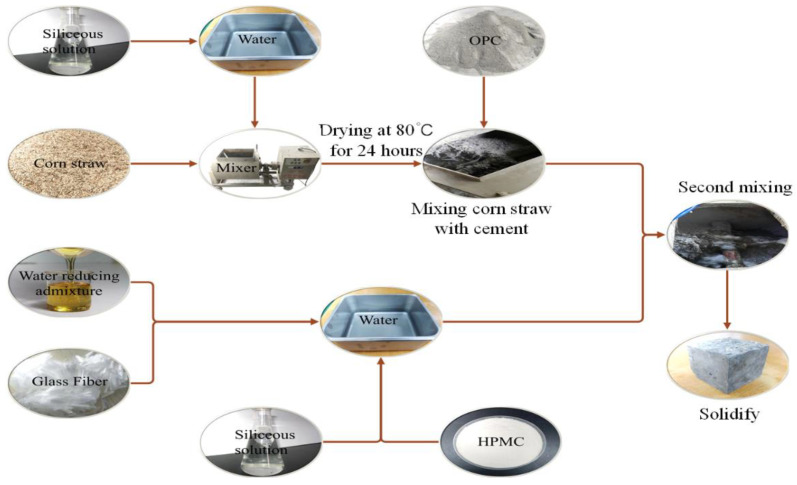
CSCC production flow chart.

**Figure 4 materials-15-03199-f004:**
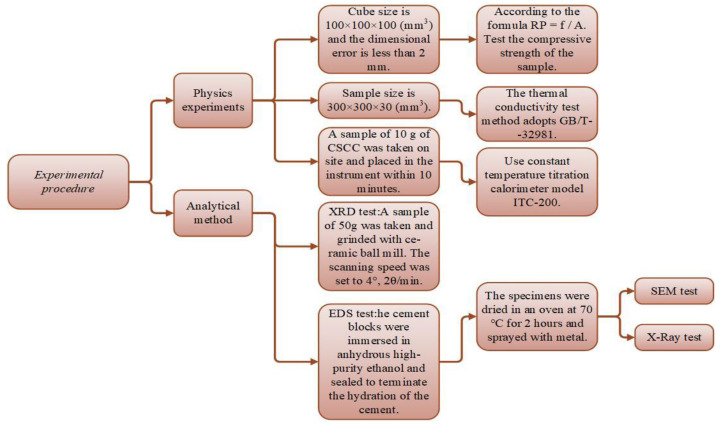
CSCC experimental procedure flow chart.

**Figure 5 materials-15-03199-f005:**
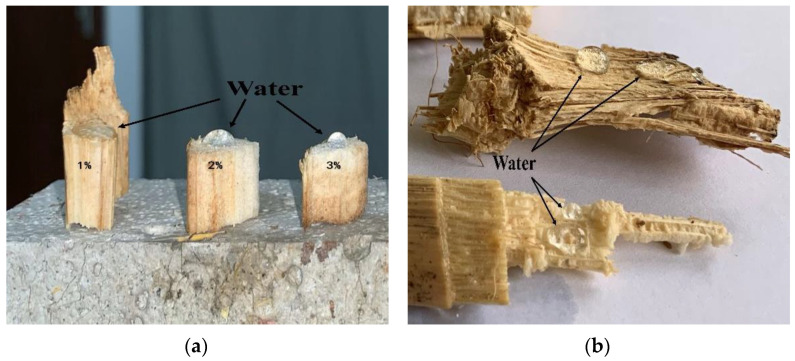
(**a**) Comparison of water absorption effect of corn straw from 1 wt.%~3 wt.%, (**b**) 3 wt.% content of silica solution concentration of corn straw produces hydrophobicity.

**Figure 6 materials-15-03199-f006:**
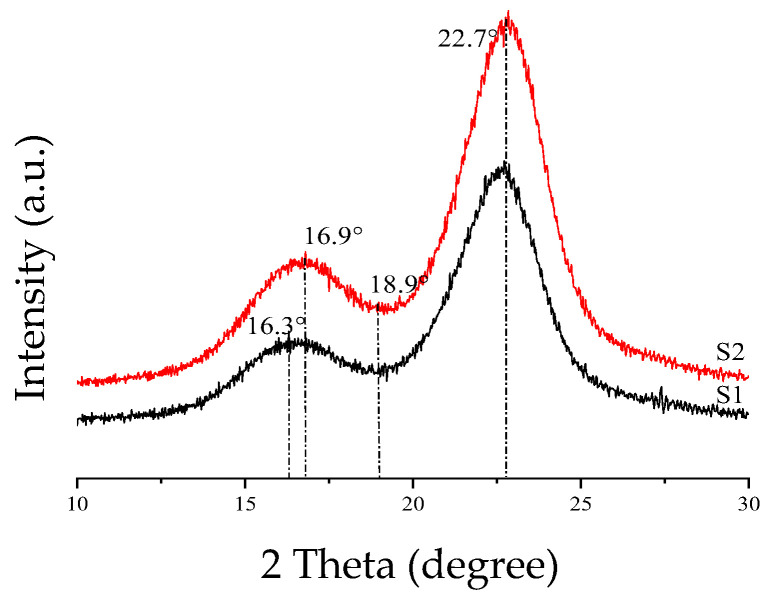
XRD patterns of corn straw: S1 corn straw, S2 corn straw after silica solution treatment.

**Figure 7 materials-15-03199-f007:**
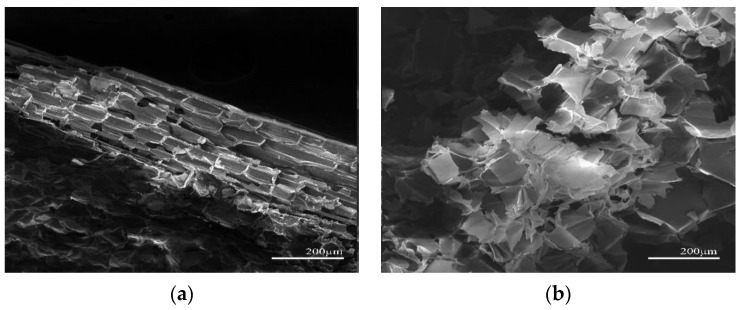
Microstructure of corn straw treated with siliceous solution: (**a**) vertical section of corn straw, (**b**) vertical section of corn straw core.

**Figure 8 materials-15-03199-f008:**
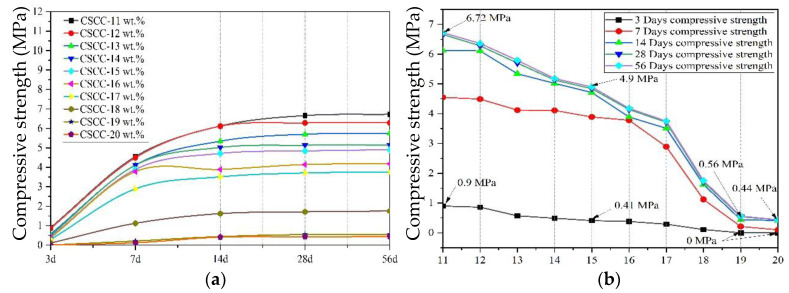
Relationship between compressive strength and age of samples. (**a**) Age of CSCC (d-days); (**b**) corn straw ratio (wt.%).

**Figure 9 materials-15-03199-f009:**
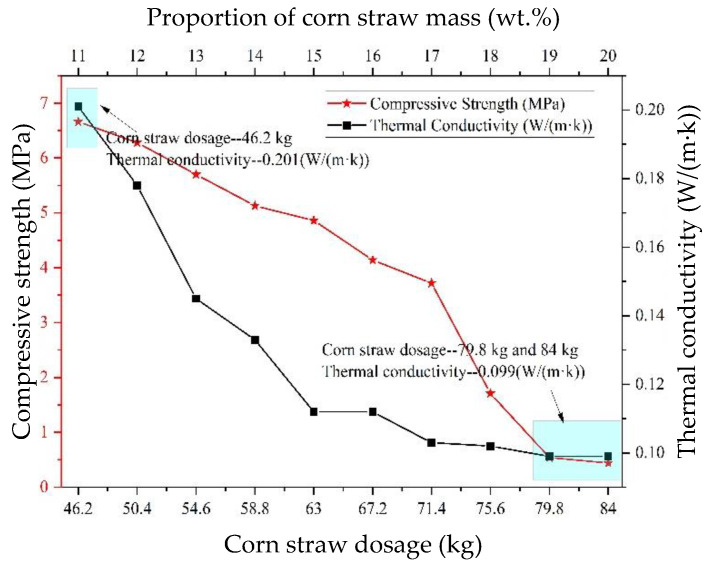
Effect of corn straw content on CSCC.

**Figure 10 materials-15-03199-f010:**
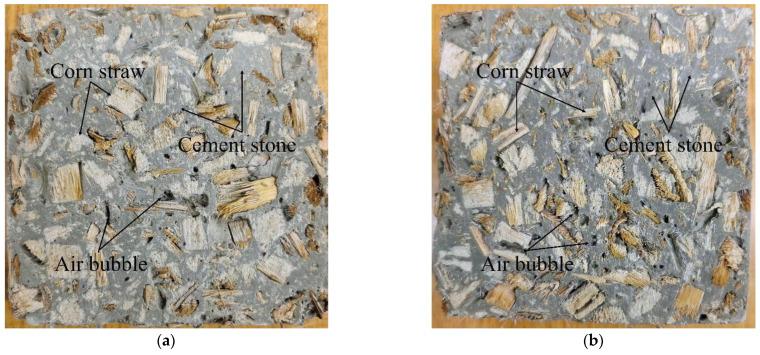
CSCC 100 × 100 × 100 mm^3^ slice: (**a**) CSCC-7, (**b**) CSCC-8.

**Figure 11 materials-15-03199-f011:**
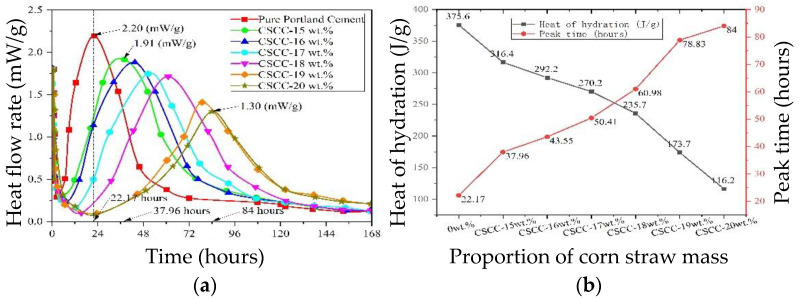
(**a**) Heat flow rate, (**b**) heat and hours of the different samples.

**Figure 12 materials-15-03199-f012:**
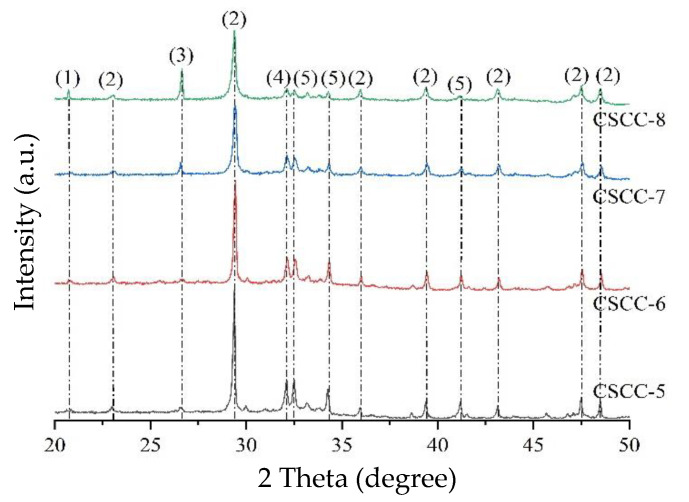
Fifty-six day XRD pattern: (1) gypsum, (2) Portlandite, (3) quartz, (4) calcium magnesium aluminum oxide silicate, (5) calcite silicate.

**Figure 13 materials-15-03199-f013:**
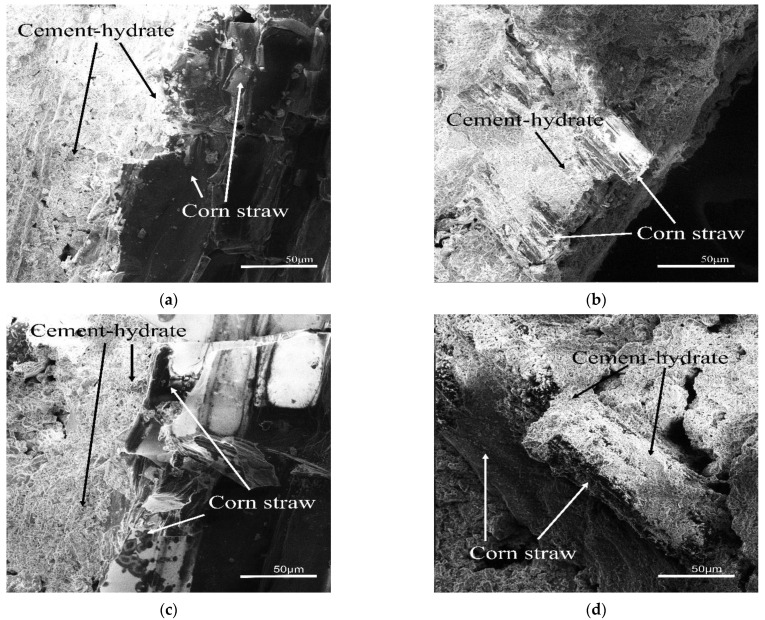
Microstructure of CSCC; (**a**) CSCC-5, (**b**) CSCC-6, (**c**) CSCC-7 and (**d**) CSCC-8.

**Figure 14 materials-15-03199-f014:**
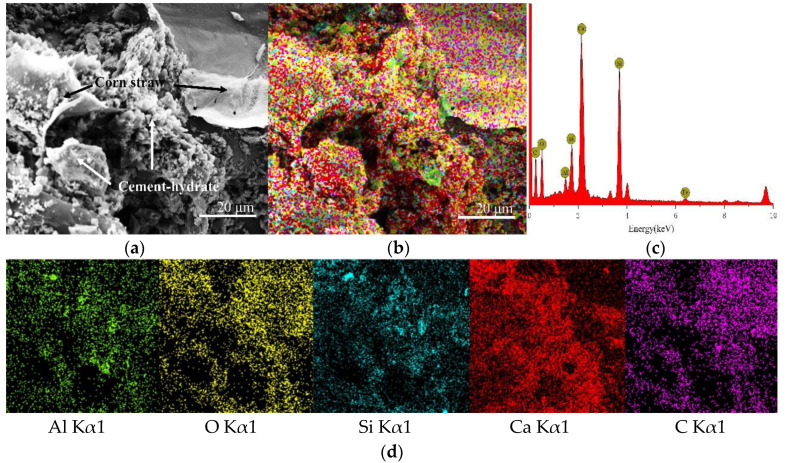
EDS scanning diagram of CSCC-5: (**a**) SEM of CSCC, (**b**) element distribution of CSCC, (**c**) energy spectrum of CSCC, (**d**) single element distribution diagram.

**Figure 15 materials-15-03199-f015:**
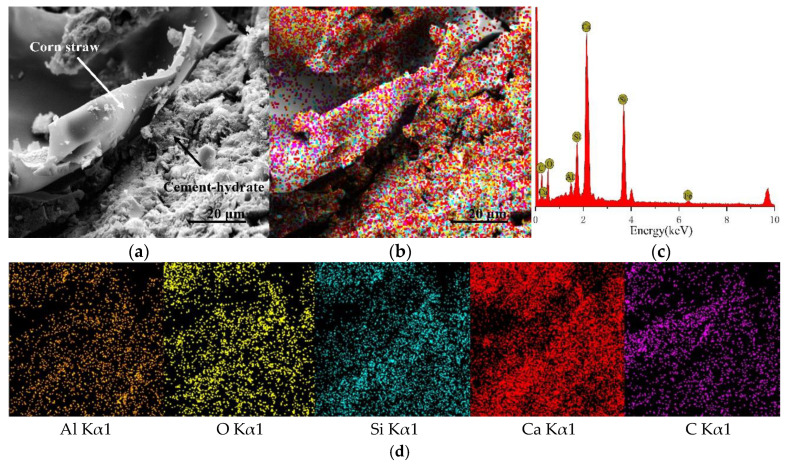
EDS scanning diagram of CSCC-5: (**a**) SEM of CSCC, (**b**) element distribution of CSCC, (**c**) energy spectrum of CSCC, (**d**) single element distribution diagram.

**Figure 16 materials-15-03199-f016:**
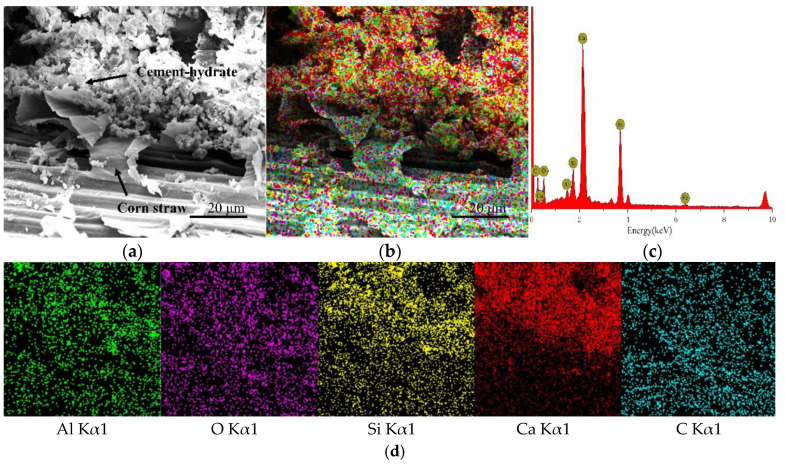
EDS scanning diagram of CSCC-5: (**a**) SEM of CSCC, (**b**) element distribution of CSCC, (**c**) energy spectrum of CSCC, (**d**) single element distribution diagram.

**Figure 17 materials-15-03199-f017:**
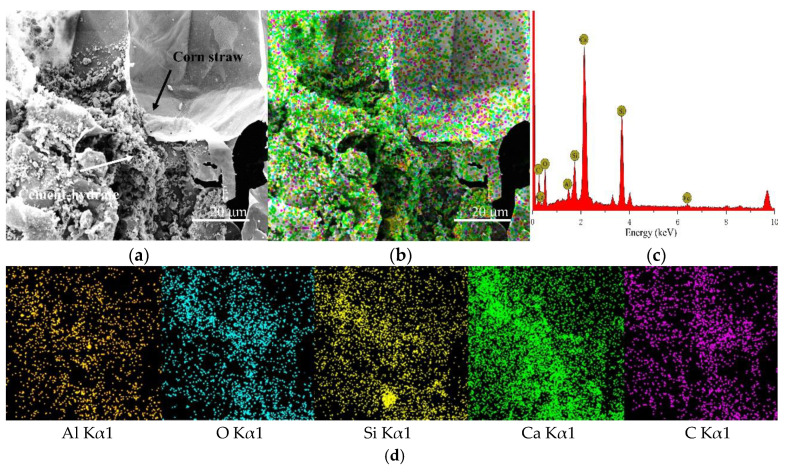
EDS scanning diagram of CSCC-5: (**a**) SEM of CSCC, (**b**) element distribution of CSCC, (**c**) energy spectrum of CSCC, (**d**) single element distribution diagram.

**Table 1 materials-15-03199-t001:** Cement main components.

Components (wt.%)	SiO_2_	Al_2_O_3_	Fe_x_O_y_	MgO	CaO	SO_3_
Quantity	22.14	5.01	5.11	1.10	1.78	2.30

**Table 2 materials-15-03199-t002:** Main components of corn straw.

Components (wt.%)	Ash	Cold Water Extract	Hot Water Extract	1% NaOH Extract	Cellulose	Lignin	Cellulose Sugar
Corn Straw	5.12	12.61	20.56	46.78	33.32	19.47	23.58

**Table 3 materials-15-03199-t003:** Main components of siliceous solution.

	Appearance	Water (wt.%)	Solids (wt.%)	Silicone (wt.%)	Specific Gravity (25 °C)	PH Value	Flammability
Siliceous Solution	Colourless	42	40	18	1.1	13	No

**Table 4 materials-15-03199-t004:** Mix formulation of the composite samples with corn straw.

Number	Corn Straw		Glass Fiber (kg)	Water Reducing Admixture (kg)	Cement (kg)	Siliceous Solution (kg)	Water to Binder (kg)	
	Percentages (wt.%)	Dosage (kg)					Water	HPMC
CSCC-1	11	46.2	0.42	2.1	420	8.4	210	0.63
CSCC-2	12	50.4	0.42	2.1	420	8.4	210	0.63
CSCC-3	13	54.6	0.42	2.1	420	8.4	210	0.63
CSCC-4	14	58.8	0.42	2.1	420	8.4	210	0.63
CSCC-5	15	63	0.42	2.1	420	8.4	210	0.63
CSCC-6	16	67.2	0.42	2.1	420	8.4	210	0.63
CSCC-7	17	71.4	0.42	2.1	420	8.4	210	0.63
CSCC-8	18	75.6	0.42	2.1	420	8.4	210	0.63
CSCC-9	19	79.8	0.42	2.1	420	8.4	210	0.63
CSCC-10	20	84	0.42	2.1	420	8.4	210	0.63

**Table 5 materials-15-03199-t005:** Physical properties of corn straw.

Corn Straw	Density (kg/m^3^)	Thermal Conductivity (W/(m·k))	The 24 Hours Water Absorption of Corn Straw (wt.%)	
			Natural	Processed
This study	488 ± 15	0.099	140	54
Reference [24]	450	0.096 ± 0.001	130~150	---

**Table 6 materials-15-03199-t006:** Compressive strength and thermal conductivity of CSCC.

Number	100 × 100 × 100 mm3 Cube Strength (MPa)					Apparent Density (kg/m^3^)	The 56 Days Thermal Conductivity (W/(m·k))
	3 Days	7 Days	14 Days	28 Days	56 Days		
CSCC-1	0.9	4.55	6.12	6.66	6.72	1058.8	0.201
CSCC-2	0.86	4.49	6.11	6.28	6.36	973.7	0.178
CSCC-3	0.57	4.12	5.34	5.7	5.79	950.3	0.145
CSCC-4	0.49	4.11	5.02	5.13	5.18	897.4	0.133
CSCC-5	0.41	3.89	4.71	4.84	4.9	758.7	0.112
CSCC-6	0.38	3.78	3.89	4.14	4.18	664.3	0.112
CSCC-7	0.29	2.89	3.51	3.71	3.75	632.5	0.103
CSCC-8	0.11	1.12	1.62	1.71	1.75	563.4	0.102
CSCC-9	0	0.21	0.44	0.54	0.56	479.7	0.099
CSCC-10	0	0.10	0.41	0.43	0.44	410.5	0.099

**Table 7 materials-15-03199-t007:** Heat flow rate and heat of hydration of CSCC under different quantities.

	Heat Flow Rate		Heat of Hydration (J/g)
	Time (h)	Max. Peak (mW/g)	
Pure Portland Cement	22.17	2.20	375.6
CSCC-15 wt.%	37.96	1.91	316.4
CSCC-16 wt.%	43.55	1.88	292.2
CSCC-17 wt.%	50.41	1.75	270.2
CSCC-18 wt.%	60.98	1.72	235.7
CSCC-19 wt.%	78.83	1.41	173.7
CSCC-20 wt.%	84	1.30	116.2

## Data Availability

Not applicable.

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
