# Peer review of "Method for Manufacturing Corn Straw Cement-Based Composite and Its Physical Properties"

_materials, 2022, doi:10.3390/ma15093199_

Round 1

Reviewer 1 Report

The title does not correspond to the text of the paper. Correct

Add few more sentances in abstract.

The procedure for making corn straw and cement composites is not clear. Explain it

There is to many results and it's little hard to follow , maybe to split on two articles

Add a couple of recent references with a similar topic published in the journal you are applying for

Reviewer 2 Report

The manuscript entitled "Analysis of preparation and physical properties of corn straw cement-based composite walls for rural buildings" presents an interesting experimental study conducted on the obtaining of concrete with corn straw addition. However, the paper has a few issues that must be addressed. The paper needs minor revisions before it is processed further, some comments follow:

  • Table 1 - two types of iron oxides have been detected in this type of material, therefore, please replace Fe2O3 with FexOy or provide the scientific proof to support your results. Moreover, the XRD analysis from figure 12 doesn’t show the presence of phases with Fe content.
  • Table 3 – What’s Proportion stands for?
  • Figure 1 – a) Please remove it doesn’t have any scientific value, also, it doesn’t contain information necessary to assure the experiments' repeatability. b) please introduce a scale bar.
  • Line 184: please replace "and age of corn straw" with "and age of samples".
  • Figure 9 - Please introduce figure labels to highlight the areas of interest for the readers.

General remark: Figure 9 shows that the corn straw "particles" are different both in size and geometry. Therefore, without a clear particle size distribution of the corn straw, the experiment's repeatability is hard to assure. For future studies, please use corn straw particles of the same size and geometry. (long fibers will exhibit different performance than short ones, therefore, if the reinforcing particles are different it cannot be stated how the amount of particles influenced the properties because it cannot be observed what factor influenced the performances of the material - the amount of particles or the geometry of particles?!

Reviewer 3 Report

The paper "Analysis of preparation and physical properties of corn straw cement-based composite for rural buildings" presents an important alternative to the use of agroindustrial walls for the development of new building materials in cementitious materials. This role should be considered for publications, however authors should make important additional corrections:

(1) In the abstract, the authors must state more clearly and objectively which mixtures were used, in addition, a description of the main methods used is lacking.
(2) The surface treatment process in agricultural products causes several problems in cementitious matrices, the authors should show and highlight new and other studies in the literature on this topic, forming a table with some quantitative results. I suggest reading and introducing some papers, such as: 10.3390/app11073036; 10.1016/j.cscm.2022.e00968; 10.1016/j.cscm.2021.e00833.
(3) At the end of the introduction, authors should address and declare the innovation of this research. What does it really bring innovation within literature??
(4) An experimental flowchart must be added in the materials section. There are numerous results shown in the methodology section, this is not correct!
(5) The limit values ​​(maximum and minimum) of the mechanical resistance in the literature should be better stated by the authors.
(6) Discussion of micrograph results are not adequate. Authors should more clearly state the issues addressed in this research.
(7) An exploratory analysis of durability may be relevant in the discussions presented.
(8) The conclusion should address issues of challenges that this experimental program encountered in its execution.

Round 2

Reviewer 3 Report

ok.